# Adaptation and Dissemination of Korean Medicine Clinical Practice Guidelines for Traffic Injuries

**DOI:** 10.3390/healthcare10071166

**Published:** 2022-06-22

**Authors:** Kyeong-Tae Lim, Hyun-Tae Kim, Eui-Hyoung Hwang, Man-Suk Hwang, In Heo, Sun-Young Park, Jae-Heung Cho, Koh-Woon Kim, In-Hyuk Ha, Me-riong Kim, Kyoung-Sun Park, Hyoung Won Kang, Jun-Hwan Lee, Byung-Cheul Shin

**Affiliations:** 1School of Korean Medicine, Pusan National University, Yangsan 50612, Korea; kyeongtae.lim@gmail.com (K.-T.L.); hyeontae1308@hanmail.net (H.-T.K.); taichi@pusan.ac.kr (E.-H.H.); hwangmansuk@pusan.ac.kr (M.-S.H.); drheoin@pusan.ac.kr (I.H.); shl0305@gmail.com (S.-Y.P.); 2Department of Korean Medicine Rehabilitation, Pusan National University Korean Medicine Hospital, Yangsan 50612, Korea; 3Department of Korean Medicine Rehabilitation, College of Korean Medicine, Kyung Hee University, Seoul 02447, Korea; vetkong95@hanmail.net (J.-H.C.); garson83@hanmail.net (K.-W.K.); 4Jaseng Spine and Joint Research Institute, Jaseng Medical Foundation, Seoul 06110, Korea; hanihata@jaseng.org (I.-H.H.); mrk@jaseng.co.kr (M.-r.K.); 5Jaseng Hospital of Korean Medicine, Gangnam-gu, Seoul 06110, Korea; lovepks0116@jaseng.co.kr; 6Department of Neuropsychiatry, College of Oriental Medicine, Wonkwang University, Iksan 54538, Korea; dskhw@wku.ac.kr; 7KM Science Research Division, Korea Institute of Oriental Medicine, Daejeon 34054, Korea; omdjun@kiom.re.kr; 8Korean Medicine Life Science, Campus of Korea Institute of Oriental Medicine, University of Science & Technology (UST), Daejeon 34054, Korea

**Keywords:** traffic injury, clinical practice guideline, adaptation, dissemination, implementation, Korean medicine

## Abstract

In South Korea, car insurance that includes medical coverage of traditional Korean medicine (TKM) has increased exponentially. Clinical practice guidelines (CPG) for traffic injuries were established in 2016. We aimed to revise and update de novo CPG and distribute the adapted CPG to TKM practitioners and patients. Clinical key questions from previous CPG were identified and updated regarding the grade of recommendation and level of evidence using additional evidence from the literature obtained through a systematic search and the use of the Grading of Recommendations Assessment, Development, and Evaluation methodology. The dissemination and implementation of the updated CPG were conducted at the CPG Center of Korean Medicine. Ultimately, 25 recommendations based on 13 clinical key questions were developed: 2 for diagnosis, 22 for TKM treatments, and 1 for prognosis. After recognition by professional societies and certification by the CPG Center of Korean Medicine, leaflets, card news, and infographics for TKM doctors in South Korea were produced and distributed. These are the only TKM CPG for patients who have experienced traffic injuries. They are expected to contribute to standardized and evidence-based treatment using TKM and similar interventions. Moreover, disseminating the adapted CPG will promote treatment reliability and strengthen insurance coverage.

## 1. Introduction

The Global Status Report on Road Safety 2018 launched by the World Health Organization in 2018 indicated that approximately 1.35 million people die yearly because of traffic injuries and 50 million suffer non-fatal injuries [1]. Traffic injuries cause considerable economic losses for individuals, families, and nations. These losses arise from the cost of treatment and lost productivity for those killed or disabled by injuries and family members who need to take time off work or school to care for the injured. Traffic accidents cost up to the amount of 3% of the gross domestic product in most countries [2].

In South Korea, the average number of traffic accidents during the past 5 years was approximately 210,000, thereby increasing the overall auto insurance cost to USD 20 billion annually by 2020 [3,4]. The number of patients presenting to medical institutions that perform traditional Korean medicine (TKM) increased from 1,329,836 in 2019 to 1,427,734 in 2020, which is a 7.3% increase; this number is nearly double that observed in 2016. In 2020, auto insurance medical coverage payments to the TKM sector of Korean medicine hospitals and clinics were KRW 1.124 trillion (approximately USD 9.4 billion), thereby accounting for nearly half of all auto insurance medical coverage expenses [4]. This is a two-fold increase from 2016, accounting for 26.7% of the KRW 1.658 trillion in total auto insurance medical coverage posts in South Korea. These costs have recently increased exponentially [5].

TKM refers to traditional approaches to medicine practiced in South Korea and includes acupuncture, moxibustion, Chuna manual therapy (CMT), pharmacopuncture, cupping, herbal medicine, and physiotherapy [6]. Korean medicine doctors have been providing medical services without clinical practice guidelines (CPG). Standardized medical services are required because of the increasing number of patients with traffic injuries visiting medical institutions for TKM and medical coverage costs [5,7]. Accordingly, the Korean government distributed a manual to the National Evidence-based Healthcare Collaborating Agency and the Guidelines Center of Korean Medicine (GKoM) to initiate the development and implementation of de novo CPG for the treatment of traffic injuries in 2016 [8]. Subsequently, CPG for the treatment of traffic injuries were published in 2018, and a second edition was published in 2020 through an acceptance adaptation process that included formally deriving changed grades of recommendation (GoRs) and levels of evidence (LoEs), thereby synthesizing the latest domestic and international evidence to develop recommendations through expert consensus to ensure a level of methodological rigor meeting international standards and including widely used pathways in clinical settings based on objective evidence and clinician expertise.

We aimed to present updated previous CPG as well as the contents of the dissemination including standardized and effective TKM treatment. These CPG are the most systematic and standardized TKM CPG developed for traffic injury. Although these CPG are based on South Korea’s medical system, they have value as a reference for CPG development in other countries using similar interventions including acupuncture and moxibustion. Additionally, the development and implementation of CPG reduce unnecessary medical costs and increase the return-to-work rate after traffic injuries, ultimately contributing to improvements in the quality of TKM practice.

## 2. Materials and Methods

### 2.1. Overall Development Process

After de novo development of CPG [9], the GKoM at the National Development Institute of Korean Medicine (NIKOM) suggested that an assessment should be conducted by external reviewers and that changes should be made to the previously suggested GoRs and LoEs. Additionally, key clinical questions (CQs) related to previous de novo CPG were identified and updated through additional literature searches, thereby reflecting the results of online surveys of Korean medicine doctors, qualitative studies, and clinical studies conducted to establish evidence for CPG during the development period. After the guidelines were revised and updated, numerous pamphlets and brochures were distributed to Korean medicine doctors in South Korea. The overall process is summarized in Figure 1.

### 2.2. Committee Organization

The development committee was led by professors of Korean medicine rehabilitation who consulted clinical experts in each field and completed the entire process of CQ development, performed a systematic search for evidence, an analysis of the quality of evidence, GoRs, and revisions of the CPG for traffic injuries. A review and advisory committee consisting of external experts in clinical, academic, and methodological studies examined whether the clinical status in South Korea was reflected in the developed CQs and literature evidence to ensure that the LoE and GoR assignment process was correctly performed. Appendix A lists the committee members and their conflicts of interest.

### 2.3. Reorganizing the Clinical Questions

After the development of the CQs associated with the previous CPG in 2016, the results of an evaluation performed by external reviewers and additional literature evidence obtained from a systematic search revealed that the previous CQs were vaguely described, thus necessitating reorganization and segmentation. Hence, the existing CQ section focusing on patients and interventions was preserved; however, the section focused on pain was subdivided, and the evidence was updated based on further literature and systematic research findings from February 2020 to April 2020. Additional surveys, qualitative research, and an economic evaluation have been conducted to obtain sufficient evidence during the CPG development phase [10,11,12]. The development committee also studied other CPG related to traffic injuries, recommendations comparable to Korean medicine therapeutic interventions, LoEs, GoRs, and descriptions of the differences between the previous CPG found in the literature. The detailed literature search process and information on search strategy including databases, keywords, article language, and inclusion and exclusion criteria had already been published [9].

### 2.4. Changes in LoEs and GoRs and Agreement with the Delphi Method

The LoEs and GoRs of the guidelines were evaluated following the Grading of Recommendations Assessment, Development, and Evaluation (GRADE) approach developed by Cochrane’s GRADE Working Group [13]. Each identified CQ was classified as one of five categories (high, moderate, low, very low, and classical text-based) instead of as one of four categories (high, moderate, low, and insufficient), and a classical text-based level was assigned a grade corresponding to a good practice point (Table 1). According to the agreement of the development committee, CQs exhibiting a low LoE and low GoR in combination with apparent benefits and high utilization at clinical sites were re-examined. Furthermore, the updated CQs were validated by a review and advisory committee using the RAND–UCLA Appropriateness Method [14]. The committee noted whether they agreed with the LoEs and GoRs on the reorganized CQs in November 2020.

### 2.5. External Review

The revision that incorporated the external reviewers’ feedback was made after the external review of the draft CPG for traffic injuries by the GKoM in October 2020. An external evaluation was conducted by 11 members of the GKoM committee and the external monitoring committee, including methodology and CPG development experts, to ensure that the CPG modification plan was complete and followed the aforementioned process.

### 2.6. Recognition by Professional Societies

The CPG for traffic injuries were approved by six related societies: the Society of Korean Medicine Rehabilitation, the Korean Acupuncture and Moxibustion Medicine Society, the Korean Pharmacopuncture Institute, the Korean Society of Chuna Manual Medicine for Spine and Nerves, the Korean Society of Oriental Neuropsychiatry, and the Society of Sports Korean Medicine. To obtain academic approval, professional societies reviewed the academic and external feasibility, strict development, content validity, applicability, and the feasibility of recommendations. Academic certification results were submitted to the GKoM.

### 2.7. Certification

The certification of CPG for traffic injuries was evaluated by a special review and assessment committee formed by GKoM members. After a systematic review of the South Korean medicine clinical procedures exhibiting methodological rigor and external validity, the CPG for traffic injuries obtained final certification from the GKoM under the auspices of NIKOM in 2020.

### 2.8. Dissemination and Implementation of the CPG

The developing committee produced pamphlets, card news, and infographics regarding standardized Korean medicine treatment to increase the use of CPG for traffic injuries at clinical sites by both Korean medicine doctors and patients.

## 3. Results

After the reorganization process, the revised recommendations increased from 13 to 25 [9]. Each recommendation was divided into sections based on additional literature research and surveys, qualitative research, and economic evaluation studies. The LoEs and GoRs were updated, yielding the CQs and recommendations presented in Table 2.

### 3.1. Diagnosis

#### 3.1.1. Collaborative Medical Treatment Using TKM and Conventional Medicine

Recommendation 1: Collaborative medical treatment may be considered for patients of all ages suspected of having whiplash-associated disorder (WAD) III and WAD IV after a physical examination (GoR/LoE: C/very low). A survey of medical collaboration using TKM and conventional medicine in 2019 indicated that 80.5% of patients with traffic injuries thought that collaboration was necessary to enhance their treatment outcomes [15]. South Korea has a dualized healthcare system, and a grade C recommendation was assigned based on the clinical experience and survey research conducted by the CPG development committee because TKM doctors can overcome potential limitations when treating patients with traffic injuries by collaborating with conventional medicine doctors.

#### 3.1.2. Syndrome Pattern

Recommendation 2: Blood stagnation syndrome is the first pattern to occur after traffic injury for patients of all ages. Depending on the patient’s condition or the doctor’s judgment, more detailed pattern identification according to qi–blood, viscera–entrails, cold–heat, and yin–yang may be considered (GoR/LoE: C/very low). The frequency of the classification of TKM pattern syndrome according to two questionnaire surveys of Korean medicine doctors has been presented previously [10,11]. TKM treatment for pattern syndromes is highly effective for patients with traffic injuries in clinical situations; therefore, a grade C recommendation was assigned based on the literature and the clinical experience of the CPG development committee.

### 3.2. Treatment

#### 3.2.1. Acupuncture (Electroacupuncture, Motion-Style Acupuncture Technique)

Recommendation 3-1: A combination of usual care and acupuncture treatment should be considered instead of other usual care interventions for the improvement of neck pain experienced by patients with WAD I and WAD II (aged 19–70 years) (GoR/LoE: B/moderate). There was a statistically significant difference in pain (standardized mean difference [SMD], 1.43; 95% confidence interval [CI]: 0.51, 2.35) in the control group who received concurrent sham acupuncture and usual care and pain of the group that received usual care alone; however, there was no significant difference in disability (SMD, 0.31; 95% CI: −0.07, 0.69) [16,17,18]. No significant adverse events were reported; however, minor adverse events were reported, indicating that the benefit outweighed the risk of safety concerns. Acupuncture is the most common treatment used in South Korea, and its benefits may be trusted; therefore, the development committee rated it as grade B.

Recommendation 3-2: The combination of usual care with acupuncture treatment may be considered for low back pain and function improvement in patients with WAD I and WAD II (aged 19–70 years) (GoR/LoE: C/very low). Acupuncture and electroacupuncture are common treatments for low back pain and function improvement in patients with traffic injuries in South Korea. Because no randomized controlled trials focused on the treatment of low back pain and disability with acupuncture, the LoE was rated very low. Based on the literature findings and clinical experience of the CPG development committee, it was assigned a grade of C.

Recommendation 3-3: A combination of usual care and electroacupuncture treatment should be considered instead of other usual care interventions to improve neck pain experienced by patients with WAD I and WAD II (aged 19–70 years) (GoR/LoE: B/moderate). Concurrent treatment with electroacupuncture and usual care substantially reduced neck pain compared with sham electroacupuncture or usual care for adults (SMD, 0.53; 95% CI: 0.23, 0.84); however, there were no significant differences in the disability index (SMD, 0.09; 95% CI: −0.22, 0.39) [19,20]. Several studies have not reported serious adverse events. Only minor side effects such as temporary pain have been observed; therefore, there has been no cause for concern regarding its safety, and the benefits outweighed the harms. Because electroacupuncture is more frequently used for musculoskeletal patients when they visit a Korean medicine institution, the CPG development committee proposed a GoR of B.

Recommendation 3-4: The combination of usual care with additional electroacupuncture treatment may be considered for symptom improvement of low back pain for patients with WAD I and WAD II (aged 19–70 years) (GoR/LoE: C/low). The combination of usual care with additional electroacupuncture treatment did not provide significant improvement of the pain (SMD, 0.31; 95% CI: −0.20, 0.82) or disability index (SMD, 0.29; 95% CI: −0.22, 0.80) [21]. The development committee assigned a low LoE based on the number of participants in one study. However, the CPG development committee assigned a GoR of C based on the clinical use of electroacupuncture.

Recommendation 3-5: The combination of usual care with the motion-style acupuncture technique (MSAT) should be considered instead of other usual care interventions to improve neck pain experienced by patients with WAD I and WAD II (aged 19–70 years) (GoR/LoE: B/moderate). The MSAT is a TKM technique that reduces musculoskeletal pain by combining breathing exercises and acupuncture treatment [22]. For patients with traffic injuries, the MSAT and usual care considerably relieved pain intensity compared to usual care alone (SMD, 0.85; 95% CI: 0.44, 1.27); however, there were no significant differences in the disability index (SMD, 0.29; 95% CI: −0.11, 0.69) [23]. The combination of MSAT with usual care for patients with traffic injuries only poses minor low-level concerns, and the clinical benefits were reliable. Therefore, the CPG development committee assigned a GoR of B.

#### 3.2.2. Pharmacopuncture

Recommendation 4-1: Pharmacopuncture may be considered instead of acupuncture treatment alone for the alleviation of neck and low back pain experienced by patients with WAD I and WAD II (aged 19–70 years) (GoR/LoE: C/very low). Pharmacopuncture combines the principles of meridians and pharmacotherapy and is widely used at TKM institutions. No clinical study found that pharmacopuncture has a better therapeutic effect on neck pain or low back pain after traffic injuries than usual care or sham pharmacopuncture alone. However, because it is currently included in the medical coverage provided by auto insurance in South Korea, the CPG development committee assigned a GoR of C based on clinical experience and the literature.

Recommendation 4-2: The combination of pharmacopuncture with usual care instead of acupuncture treatment alone should be considered for the improvement of neck pain for patients with WAD I and WAD II (aged 19–70 years) (GoR/LoE: B/moderate). Three trials that compared pharmacopuncture and usual care with acupuncture treatment alone for neck pain after traffic accidents included pain intensity and disability index assessments. Three studies found statistically significant differences in pain intensity when pharmacopuncture was used in combination with usual care (SMD, −0.90; 95% CI: −1.29, −0.51) [24,25,26]. One study that included the disability index revealed a significant difference (SMD, −0.98; 95% CI: −1.92, −0.04). The LoE was rated as moderate because only minor adverse events were reported in the three studies, implying that pharmacopuncture benefits outweighed the harm. Consequently, the CPG development committee assigned it a GoR of B.

Recommendation 4-3: The combination of pharmacopuncture with usual care may be considered for the improvement of low back pain experienced by patients with WAD I and WAD II (aged 19–70 years) (GoR/LoE: C/very low). The LoE was rated as very low because it included only one study. Even though a small number of trials failed to find statistically significant effects on the pain intensity (SMD, 1.23; 95% CI: −0.40, −2.86) and disability index (SMD, 0.58; 95% CI: −0.11, 1.27), concurrent pharmacopuncture treatment was frequently used for low back pain alleviation and function recovery after traffic accidents. The CPG development committee assigned a GoR of C [27].

#### 3.2.3. Chuna Manual Therapy

Recommendation 5-1: CMT alone may be considered to alleviate neck pain and improve the function of patients with WAD I and WAD II (aged 19–70 years) (GoR/LoE: C/very low). CMT is popular in South Korea, where other manipulation methods, such as osteopathy, chiropractic, and other manipulative therapies, were developed [28]. However, there is no recognized sham CMT treatment, and identifying relevant clinical data is challenging. Because no literature has supported CQ, the CPG development committee set the LoE as low and assigned a GoR of C based on clinical experience and the literature. There is no evidence indicating that CMT alone is efficacious.

Recommendation 5-2: The combination of CMT with usual care may be considered for the improvement of neck pain for patients with WAD I and WAD II (aged 19–70 years) (GoR/LoE: C/low). When a control group who received concurrent CMT with usual care was compared with a group who received usual care alone, there was no statistically significant difference in pain (SMD, 0.48; 95% CI: 0.01, 0.94) or disability (SMD, 0.48; 95% CI: 0.02, 0.94) [29,30]. However, the studies were small-scale and, because of the nature of CMT, blinding was not successful. Hence, the LoE was evaluated as low. Adverse events were reported by one study; however, they were minor adverse events. Therefore, CMT for neck pain experienced by patients with traffic injuries was declared safe, and its benefits were considered to outweigh the harm. Additionally, since April 2019, the cost of CMT has been reimbursed by the National Health Insurance Program of South Korea, and its use at clinical sites has expanded. As a result, the CQ received a GoR of C from the CPG development committee.

Recommendation 5-3: A combination of CMT with usual care may be considered for the improvement of low back pain experienced by patients with WAD I and WAD II (aged 19–70 years) (GoR/LoE: C/low). One study found a significant difference in the pain index when comparing patients with back pain induced by traffic accidents treated with CMT plus usual care and patients treated with usual care alone (SMD, −3.30; 95% CI: −4.74, −1.95) [31]. No serious adverse events were reported, and only minor adverse events were noted. CMT is widely used in clinical settings for patients with low back pain and is currently covered by car insurance plans in South Korea. Therefore, the CPG development committee assigned a GoR of C.

#### 3.2.4. Moxibustion

Recommendation 6-1: Moxibustion alone may be considered for the alleviation of neck and low back pain experienced by patients with WAD I and WAD II (aged 19–70 years) (GoR/LoE: C/very low). No clinical study has found that moxibustion alone has a better therapeutic effect on neck or low back pain after traffic injuries than usual care or placebo alone. Hence, the CPG development committee set the LoE as low. However, a GoR of C was assigned because moxibustion is frequently used in South Korean clinical settings to alleviate neck and low back pain intensity and function recovery.

Recommendation 6-2: The combination of moxibustion (indirect) with usual care may be considered instead of usual care alone for neck pain experienced by patients with WAD I and WAD II (aged 19–70 years) (GoR/LoE: C/low). Compared to usual care alone, routine care combined with moxibustion substantially relieves neck pain and improves the function of adult patients (pain SMD, −0.77 [95% CI: −1.21, −0.33]; disability SMD, −0.51 [95% CI: −0.94, −0.08]) [32]. A previous study did not mention any adverse events, and moxibustion treatment has not raised any safety issues. Its benefit was thought to be greater than its harm. Hence, the development committee assigned a GoR of C.

Recommendation 6-3: The combination of moxibustion (indirect) with usual care may be considered for improving low back pain experienced by patients with WAD I and WAD II (aged 19–70 years) (GoR/LoE: C/very low). No clinical study has found that concurrent treatment with moxibustion and usual care improves low back pain after traffic injuries. Despite the lack of clinical research corresponding to the CQ, the development committee assigned a GoR of C and a very low LoE based on the literature and the clinical experience of the CPG committee.

#### 3.2.5. Cupping Therapy

Recommendation 7: Cupping alone or concurrent therapy with cupping and usual care may be considered for patients with WAD I and WAD II (aged 19–70 years) to improve neck and low back pain and function (GoR/LoE: C/very low). No clinical study has found that cupping therapy alone results in improved neck and low back pain and function after traffic injuries. A GoR of C and a very low LoE were assigned based on South Korea’s clinical settings, the literature, and the clinical experience of the CPG development committee.

#### 3.2.6. Korean Medicine Physiotherapy

Recommendation 8: Korean medicine physiotherapy alone or combined with usual care may be considered for patients with WAD I and WAD II (aged 19–70 years) to improve neck and low back pain and function (GoR/LoE: C/very low). No clinical study has found that Korean medicine physiotherapy alone or in combination with usual care results in improved neck and low back pain and function after traffic injuries. A GoR of C and very low LoE were assigned based on South Korea’s clinical settings, the literature, and the clinical experience of the CPG development committee.

#### 3.2.7. Herbal Medicine

Recommendation 9: Herbal medicines alone or in combination with usual care according to pattern identification may be considered for patients with WAD I and WAD II (aged 19–70 years) to improve neck and low back pain and function (GoR/LoE: C/very low). No randomized controlled trial has shown that herbal medicine alone or in combination with usual care results in improved neck and low back pain and function after traffic injuries. The frequency of herbal medicine use has been evaluated using questionnaire surveys completed by Korean medicine doctors [10]. A GoR of C and a very low LoE were assigned based on South Korea’s clinical settings, the literature, and the clinical experience of the CPG development committee.

#### 3.2.8. Fractures

Recommendation 10: Korean medicine treatments alone or in combination with usual care may improve pain and function of patients (aged 19–70 years) with fractures caused by traffic accidents (GoR/LoE: C/very low). No large-scale clinical trials that satisfied the CQs have been performed, and only a few simple case reports have been published. TKM is frequently used in combination with usual care to treat fractures caused by traffic accidents to provide conservative treatment or rehabilitation, and better results can be expected. Therefore, the CPG development committee assigned a very low LoE and a GoR of C based on the current state of TKM and expert agreements.

#### 3.2.9. Post-Traumatic Stress Disorder

Recommendation 11: Korean medicine treatment alone or in combination with usual care may improve psychological symptoms experienced by patients (aged 19–70 years) with acute stress disorder and post-traumatic stress disorder (PTSD) after traffic accidents (GoR/LoE: C/very low). A survey of the prevalence of acute stress disorder and PTSD for patients after traffic injuries and the validity of screening tests conducted by the development committee found that the stress response after traffic accidents often leads to PTSD [33]. Furthermore, psychological and physical approaches are required for traffic injuries. A study of the correlation between Korean medicine and PTSD revealed that it is possible to screen for and diagnose PTSD with the use of questionnaire evaluations. A very low LoE and a GoR of C were assigned based on the clinical experience of the CPG development committee and the literature; however, no studies have conformed to the CQ.

#### 3.2.10. Combination of Chuna Manual Therapy and Pharmacopuncture

Recommendation 12-1: The combination of CMT and pharmacopuncture may be considered instead of CMT alone or pharmacopuncture alone to improve neck and low back pain experienced by patients with WAD I and WAD II (aged 19–70 years) (GoR/LoE: C/very low). Regarding low back and neck pain induced by a traffic accident, no randomized, controlled trials have compared the concurrent treatment of CMT and pharmacopuncture to CMT alone and pharmacopuncture alone. According to the results of a survey of Korean medicine doctors conducted in 2016, CMT and pharmacopuncture were often used simultaneously, and 58.4% of the respondents reported that concurrent CMT and pharmacopuncture were not covered by insurance in South Korea. Although no study matching the CQ was found, a very low LoE and a GoR of C were assigned based on the literature and the clinical experience of the CPG development committee.

Recommendation 12-2: The combination of pharmacopuncture with usual care (conventional Korean medicine treatment including CMT) may be considered instead of usual care to improve neck pain experienced by patients with WAD I and WAD II (aged 19–70 years) (GoR/LoE: C/low). One study reported that compared to usual care alone, the combination of pharmacopuncture with usual care substantially relieved neck pain (pain SMD, 1.21 [95% CI: 0.67, 1.75]; disability SMD, 0.79 [95% CI: 0.27, 1.30]) [34]. Because the study did not mention any adverse events, concurrent treatment did not raise any safety issues, and the benefits were thought to be greater than the harm. Therefore, the CPG development committee assigned a GoR of C.

Recommendation 12-3: The combination of CMT with usual care (conventional Korean medicine treatment including pharmacopuncture) should be considered instead of usual care alone for improving neck pain experienced by patients with WAD I and WAD II (aged 19–70 years) (GoR/LoE: B/moderate). Compared with usual care alone, concurrent CMT with usual care substantially relieves neck pain (pain SMD, 0.87; [95% CI: 0.57, 1.17]; disability SMD, 0.62 [95% CI: 0.32, 0.91]) [34]. When the development committee evaluated the adverse events recorded, they concluded that the concurrent treatment poses low-level safety concerns because minor, but not severe, adverse events were observed. CMT and pharmacopuncture are frequently used together. This cost-effective combination is more efficient for acquiring incremental quality-adjusted life-years from social and health management standpoints when treating traffic injuries [12]. According to the expert consensus, the CPG development committee assigned a GoR of B because the benefit was greater than the harm in the medical setting.

### 3.3. Prognosis

Recommendation 13: Korean medicine treatment alone or concurrent treatment with usual care may improve the rate of return to normal life and work for patients with traffic injuries (GoR/LoE: C/very low). Although several survey studies have been performed, no clinical trials have found matching rates of returning to life and work associated with TKM treatment alone and with concurrent treatment with usual care. According to a survey of Korean medicine doctors in 2016 and 2017 and another survey of patients with traffic injuries in 2018, patients (65.7%) and Korean medicine doctors (31.4%) indicated that the severity of the traffic accident was a factor affecting the prognosis for patients [10,11,15]. The expert consensus and results of a clinical utility survey led to a GoR of C and a very low LoE.

### 3.4. Dissemination of Clinical Practice Guidelines

The CPG development committee produced and distributed CPG infographics to the community of Korean medicine doctors during annual academic conferences. It also produced and distributed online branding card news and patient handouts to the general public in 2020 (Appendix A). Additionally, the results of ongoing research will be regularly uploaded and made available through the Association of Korean Medicine’s bulletin board and website dedicated to the development and dissemination of CPG (http://www.nckm.or.kr, accessed on 12 November 2021).

## 4. Discussion

Traffic accidents have costly consequences. Medical expenses for traffic accidents in Korea have increased to KRW 2.337 trillion (approximately USD 20 billion) and are steadily increasing. The increase in medical expenditures associated with traffic accidents has also resulted in social and economic costs. In South Korea, where a binary healthcare system exists, the number of Korean medical institutions for traffic accident patients has continuously increased since 2016, when the research and development of TKM CPG began. These institutions are responsible for half of all traffic accident medical expenses. As a result, the demand for standardized treatment was always increased as the number of traffic accident injury patients accessing Korean medical institutions grows, as does medical expenses.

There are clear limitations to applying conventional treatment when there is no clear reason for its use despite the inconvenience of pain or poor function after traffic accidents. After a traffic accident, pain and other problems can exist despite the absence of damage to the musculoskeletal system or nervous system. However, there is a limit to the application of conventional treatment for symptoms, such as dizziness and pain. Therefore, patients with traffic injuries may benefit from diagnosis and therapy using TKM, which combines the total structure and function of the body based on individual constitutional traits. Because many patients with traffic injuries are satisfied with the treatment and management provided by TKM institutions, the number of patients opting for TKM is increasing. Moreover, patients who experience pain that persists after treatment at other medical institutions often resort to TKM treatment.

Various CPGs address traffic accidents in other countries; however, in South Korea, none existed. South Korea has a unique medical system divided into conventional medicine and TKM. Although the need for standardized and systematic guidelines has steadily increased in South Korea, Korean medicine doctors consisting of methodological and clinical experts have not gathered to prepare CPG for traffic injuries that consider TKM until now. For this reason, about 50 CPG related to TKM are being developed under the guidance of the South Korean government. The CPG for traffic injuries were created to focus on the therapeutic concepts that Korean medicine doctors expect to enhance the clinical application of Korean medicine treatments on a practical level, including acupuncture, pharmacopuncture, and CMT for patients with traffic injuries. The CPG was developed based on the research methodology used for treatment guidelines, and recommendations were made based on various CQs related to traffic accidents. Compared to interventions currently used by Korean medicine doctors, more standardized treatment methods are recommended by the CPG. This is significant for creating treatment procedures and a cost-effective TKM treatment system on national and international levels. This CPG for traffic injuries have been approved by relevant professional Korean medicine societies and the GKoM, demonstrating their usefulness.

Since 2016, when the de novo development of CPG for traffic injuries using GRADE methodology began, there have been numerous recommendations with unacceptable GoRs and insufficient LoEs because of the lack of relevant medical literature. The development committee has addressed these constraints by organizing and commencing multi-center, randomized clinical trials and submitting research papers for publication. It was necessary to search for evidence in the literature, including studies other than randomized, clinical trials, to support clinical considerations before the CPG could be adapted. However, more systematic and well-designed clinical studies are required to confirm the adapted CPG.

De novo CPG for traffic injuries was approved and published in 2018, and the adapted CPG for traffic injuries were certified in 2020. However, two surveys conducted after the de novo CPG found that it was difficult to communicate and promote this CPG to TKM doctors at clinical sites. Therefore, the development committee has tried to communicate and promote the updated CPG through various domestic and international conferences, card news, infographics, and leaflets to inform TKM doctors of the systematic treatment procedures. Additionally, the following attempts were made to convey the patients’ perspectives. Qualitative and retrospective studies of the experiences of patients with traffic injuries and a search of the literature were performed. Surveys of patients visiting TKM institutions in South Korea have also been conducted to reflect patients’ opinions of the adapted CPG. Furthermore, a poll found that medical practitioners need to provide appropriate information, such as the precise diagnosis and treatment techniques to the patients; therefore, this was cited as a recommendation and clinical consideration.

This CPG has a few limitations. First, it is difficult to develop evidence-based CPG based on standardized diagnosis and treatment principles because diagnoses vary according to individual patient characteristics, constitutions, and symptom patterns. Thus, the majority of evidence is insufficient to support recommendations due to the lack of well-designed RCTs. Additionally, regional constraints, with most of the studies taking place in China or South Korea, lead to some difficulty in developing evidence-based CPG. More well-designed RCTs are required. Therefore, it was difficult to adequately reflect the clinical reality of TKM using only existing clinical research methods.

The second limitation of CPG is that conducting clinical research to generate evidence is frequently challenging because of institutional constraints. Additionally, it was not possible to objectively evaluated whether the dissemination brochures, infographics, and online branding news were sufficiently delivered to patients or TKM doctors.

One of the strengths of this study was the continuous searching and updating of evidence after the de novo development of CPG. In addition, the CPG aims to synthesize and describe research findings to best reflect the characteristics of TKM. This guideline will be recommended widely for use and provide evidence for areas that need to be supplemented through step-by-step research in the future. This CPG will assist in increasing the treatment efficacy of TKM as it is disseminated to actual clinical settings, thus increasing the clinical application of TKM.

## 5. Conclusions

We revised and updated the previous CPG for traffic injuries to include TKM and disseminated them in various ways so that they can be used by more TKM doctors. This guideline is the only CPG for patients with traffic injuries treated with TKM. The adapted CPG will be widely used in clinical practice and enhance the reliability of TKM to strengthen national insurance coverage in South Korea. We expect this CPG will contribute to all practitioners using similar interventions, including acupuncture, moxibustion, CMT, and cupping.

## Figures and Tables

**Figure 1 healthcare-10-01166-f001:**
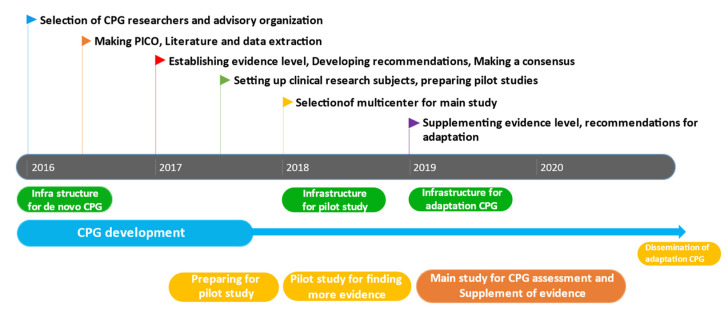
Overall development process of CPG.

**Table 1 healthcare-10-01166-t001:** Definition of the level of evidence and grade of recommendation.

Level of Evidence	Definition	
High	The authors are highly confident that the true effect is similar to the estimated effect.	
Moderate	The authors have moderate confidence in their estimated effect. The estimated effect is expected to be close to the true effect, but it may be significantly different.	
Low	The authors have a low level of confidence in the estimated effect. The true effect may be significantly different from the estimated effect.	
Very low	The authors have little confidence in the estimated effect. The true effect will be significantly different from the estimated effect.	
Classical text-based	Although few evidence-based studies using methodological approaches have been undertaken, there is evidence recorded in classical texts such as existing traditional Korean medicine books and high utilization of clinical venues.	
**Grade of Recommendation**	**Definition**	**Notation**
A	The benefits are clear and the method is highly utilized in clinical practice.	Is recommended
B	The benefits are reliable, the method has high or medium utilization in clinical practice, or the clinical benefits are clear despite insufficient research data.	Should be considered
C	The benefits are not reliable, but the method has high or medium utilization in clinical practice.	May be considered
D	The benefits are not reliable and may have detrimental consequences.	Is not recommended
GPP	There is an expert consensus-based on bibliographic evidence or utilization in clinical practice.	Is recommended based on expert consensus

GPP; good practice point.

**Table 2 healthcare-10-01166-t002:** Changes in the recommendation grade and quality of evidence for clinical key questions during adaptation periods.

Recommendation	Quality of Evidence		Grade of Recommendation	
	De Novo	Adaptation	De Novo	Adaptation
R1	GPP	C	Insufficient	Very low
R2	GPP	C	Insufficient	Very low
R3-1	B	B	Moderate	Moderate
R3-2	C	C	Low	Very low
R3-3	GPP	B	Insufficient	Moderate
R3-4	-	C	-	Low
R3-5	-	B	-	Moderate
R4-1	B	C	Moderate	Very low
R4-2	C	B	Low	Moderate
R4-3	GPP	C	Insufficient	Very low
R5-1	GPP	C	Insufficient	Very low
R5-2	GPP	C	Insufficient	Low
R5-3	-	C	-	Low
R6-1	C	C	Low	Very low
R6-2	C	C	Low	Low
R6-3	GPP	C	Insufficient	Very low
R7-1	GPP	C	Insufficient	Very low
R7-2	GPP	-	Insufficient	-
R8-1	GPP	C	Insufficient	Very low
R8-2	GPP	-	Insufficient	-
R9-1	GPP	C	Insufficient	Very low
R9-2	GPP	-	Insufficient	-
R10	GPP	C	Insufficient	Very low
R11	GPP	C	Insufficient	Very low
R12-1	GPP	C	Insufficient	Very low
R12-2	GPP	C	Insufficient	Low
R12-3	-	B	-	Moderate
R13	GPP	C	Insufficient	Very low

GPP: good practice point.

## Data Availability

Not applicable.

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
