# Peer review of "Adaptation and Dissemination of Korean Medicine Clinical Practice Guidelines for Traffic Injuries"

_healthcare, 2022, doi:10.3390/healthcare10071166_

Round 1

Reviewer 1 Report

This manuscript entitled “Adaptation and Dissemination of Korean Medicine Clinical Practice Guidelines for Traffic Injuries” primarily aimed to revise and update de novo clinical practice guidelines and distribute the adapted clinical practice guidelines to traditional Korean medicine practitioners and patients. The objectives and methods of the study are carefully presented. However, to enhance the quality of the manuscript, revise suggestions are given below.

1.      Introduction, “This is a 55 two-fold increase from 2016, accounting for 26.7% of the KRW 1.658 trillion in total auto insurance medical coverage posts in South Korea”, please add some references to support this sentence.

2.      Introduction, please use more references to support your introduction.

3.      Introduction, the novelty and value of the study should be highlighted.

4.      Methods and Results sections require more figures and tables to present.

5.      Discussion, the discussion part seems insufficient, please add more.

6.      Discussion, there are some errors in the formatting of the article, please correct it.

7.      What are the limitations of this study? Please provide relevant description.

8.      Please do check the language and grammar mistakes throughout the whole article to further improve clarity.

9.      In summary, please make sure that your manuscript is properly prepared and formatted before submitting a revision.

Author Response

Reviewer 1.

This manuscript entitled “Adaptation and Dissemination of Korean Medicine Clinical Practice Guidelines for Traffic Injuries” primarily aimed to revise and update de novo clinical practice guidelines and distribute the adapted clinical practice guidelines to traditional Korean medicine practitioners and patients. The objectives and methods of the study are carefully presented. However, to enhance the quality of the manuscript, revise suggestions are given below.

  1. Introduction, “This is a 55 two-fold increase from 2016, accounting for 26.7% of the KRW 1.658 trillion in total auto insurance medical coverage posts in South Korea”, please add some references to support this sentence.

  1. Introduction, please use more references to support your introduction.

  1. Introduction, the novelty and value of the study should be highlighted.

  1. Methods and Results sections require more figures and tables to present.

  1. Discussion, the discussion part seems insufficient, please add more.

  1. Discussion, there are some errors in the formatting of the article, please correct it.

  1. What are the limitations of this study? Please provide relevant description.

  1. Please do check the language and grammar mistakes throughout the whole article to further improve clarity.

  1. In summary, please make sure that your manuscript is properly prepared and formatted before submitting a revision.

-> Answer

Thank you for your comment. We have revised the manuscript in light of your suggestions.

1-3. We've added some references to the sentences you indicated in the introductions, and also phrases to support our study’s object.

4. We’ve added figures in methods to summarise our study’s overall process.

5-6. We've added some sentences and revised some errors in discussion.

7. This CPG has a few limitations. First, the level of evidence and grade of recommendation was set low due to the lack of well-designed clinical research which reflects the clinical reality of traditional Korean medicine. Also, regional constraints lead to some difficulty in developing evidence-based CPG due to most of the studies taking place in China or South Korea. We revised these limitations in discussion.

8-9. We will double-check the formatting and grammar before submitting a revision.

Reviewer 2 Report

“Best of all, I pray that you will be able to do well in the COVID-19 crisis. “

Thank you for giving me the opportunity to review this wonderful manuscript. I was interested in your study about the clinical practice guidelines for traffic injuries

This paper presents adpation and dissemination of Korean medicine clinical practice guideline for traffic injuries. The Grading of Recommendations, Assessment, Development and Evaluation (GRADE) was used to draft the recommendations. In total, 25 recommendations based on 13 clinical key questions were developed: 2 for diagnosis, 22 for TKM treatments, and 1 for prognosis. 

I thought this was a very thorough study and very well done.

Line 39-47           I think the author should be corrected the first paragraph in Introduction. Among the supplementary presented by the author, 'clinical algorithm for traffic injuries' seemed to be derived from 'Quebec Task Forc e on Whiplash-Associated Disorders' designed in Canada. In the case of severe injury, it seemed to be treated with Western medicine, and for after-effects after a minor or serious injury, it seems to be treated with Korean medicine alone. Therefore, I would like the author to revise the global statistics of the number of traffic accident injuries and the social cost of traffic accidents to be relevant to the content of the second paragraph.

I thought this was a very thorough study and very well done. However, I think, if the author had explained the process of adaptation and dissemination in a little more detail, it would have helped the reader to understand it better. (Mesh-Term Selection, Systematic Review, etc.)

Author Response

Reviewer 2.

“Best of all, I pray that you will be able to do well in the COVID-19 crisis. “

Thank you for giving me the opportunity to review this wonderful manuscript. I was interested in your study about the clinical practice guidelines for traffic injuries.

This paper presents adpation and dissemination of Korean medicine clinical practice guideline for traffic injuries. The Grading of Recommendations, Assessment, Development and Evaluation (GRADE) was used to draft the recommendations. In total, 25 recommendations based on 13 clinical key questions were developed: 2 for diagnosis, 22 for TKM treatments, and 1 for prognosis.

I thought this was a very thorough study and very well done.

Line 39-47 I think the author should be corrected the first paragraph in Introduction. Among the supplementary presented by the author, 'clinical algorithm for traffic injuries' seemed to be derived from 'Quebec Task Force on Whiplash-Associated Disorders' designed in Canada. In the case of severe injury, it seemed to be treated with Western medicine, and for after-effects after a minor or serious injury, it seems to be treated with Korean medicine alone. Therefore, I would like the author to revise the global statistics of the number of traffic accident injuries and the social cost of traffic accidents to be relevant to the content of the second paragraph.

I thought this was a very thorough study and very well done. However, I think, if the author had explained the process of adaptation and dissemination in a little more detail, it would have helped the reader to understand it better. (Mesh-Term Selection, Systematic Review, etc.)

-> Thank you for your kind comments and suggestion.

  1. We revised the introduction including the number of non-fatal injuries with references

  1. The detailed literature search strategy, which includes mesh-term selection and systematic review, is already included in the published De novo articles [1], as indicated by the reviewer. As a consequence of the reviewer's suggestions, we'll include the following phrase, along with the fact that the paper including the existing search results has already been published.

‘The detailed literature search process and information on search strategy including databases, keywords, article language, and inclusion and exclusion criteria had already been published’

references

[1] Kim, H.T.; Hwang, E.H.; Heo, I.; Cho, J.H.; Kim, K.W.; Ha, I.H.; Kim, M.; Kang, H.; Lee, J.; Shin, B. Clinical Practice Guidelines for the Use of Traditional Korean Medicine in the Treatment of Patients with Traffic-Related Injuries: An Evidence-Based Approach. Eur. J. Integr. Med. 2018, 18, 34–41. DOI:10.1016/j.eujim.2018.01.003.

Reviewer 3 Report

The authors revised and updated Korean medicine clinical practice guidelines for traffic injuries through a systematic literature search, review, and grading. The author's work provides valuable TKM information for traffic injuries and establishes a model for developing clinical practice guidelines. 

The authors revised and updated Korean medicine clinical practice guidelines for traffic injuries through a systematic literature search, review, and grading. The author's work provides valuable TKM information for traffic injuries and establishes a model for developing clinical practice guidelines.

However, the manuscript lacked the literature search process and necessary information, including databases, keywords, article language, inclusion and exclusion criteria, etc. Therefore, it is difficult for readers to know the author's search process from the manuscript to assess the search quality.

Author Response

Reviewer 3

The authors revised and updated Korean medicine clinical practice guidelines for traffic injuries through a systematic literature search, review, and grading. The author's work provides valuable TKM information for traffic injuries and establishes a model for developing clinical practice guidelines.

The authors revised and updated Korean medicine clinical practice guidelines for traffic injuries through a systematic literature search, review, and grading. The author's work provides valuable TKM information for traffic injuries and establishes a model for developing clinical practice guidelines.

However, the manuscript lacked the literature search process and necessary information, including databases, keywords, article language, inclusion and exclusion criteria, etc. Therefore, it is difficult for readers to know the author's search process from the manuscript to assess the search quality.

Answer

-> Thank you for your comment. Our previous article already mentioned the literature search process and necessary information on search strategy in methods. To eliminate duplication of content, just the most basic information is presented. As a consequence of the reviewer's suggestions, we'll include the following phrase in the methods, along with the fact that the paper including the existing search results has already been published.

‘The detailed literature search process and information on search strategy including databases, keywords, article language, and inclusion and exclusion criteria had already been published’

references

[1] Kim, H.T.; Hwang, E.H.; Heo, I.; Cho, J.H.; Kim, K.W.; Ha, I.H.; Kim, M.; Kang, H.; Lee, J.; Shin, B. Clinical Practice Guidelines for the Use of Traditional Korean Medicine in the Treatment of Patients with Traffic-Related Injuries: An Evidence-Based Approach. Eur. J. Integr. Med. 2018, 18, 34–41. DOI:10.1016/j.eujim.2018.01.003.
